# Green and Efficient Construction of Chromeno[3,4-*c*]pyrrole Core via Barton–Zard Reaction from 3-Nitro-2*H*-chromenes and Ethyl Isocyanoacetate

**DOI:** 10.3390/molecules27238456

**Published:** 2022-12-02

**Authors:** Ivan A. Kochnev, Alexey Y. Barkov, Nikolay S. Zimnitskiy, Vladislav Y. Korotaev, Vyacheslav Y. Sosnovskikh

**Affiliations:** Institute of Natural Sciences and Mathematics, Ural Federal University, 620000 Ekaterinburg, Russia

**Keywords:** 3-nitro-2*H*-chromenes, ethyl isocyanoacetate, Barton–Zard reaction, 2,4-dihydrochromeno[3,4-*c*]pyrroles

## Abstract

A regioselective one-pot method for the synthesis of 1-ethyl 2,4-dihydrochromene[3,4-*c*]pyrroles in 63–94% yields from available 2-phenyl-, 2-trifluoro(trichloro)methyl- or 2-phenyl-2-(trifluoromethyl)-3-nitro-2*H*-chromenes and ethyl isocyanoacetate through the Barton–Zard reaction in ethanol at reflux for 0.5 h, using K_2_CO_3_ as a base, has been developed.

## 1. Introduction

The chromenopyrrole moiety is present in a wide range of natural and synthetic compounds with a number of important useful properties [1,2,3,4,5,6,7,8,9,10,11,12,13,14,15]. Thus, chromeno[3,4-*b*]pyrrole scaffold is the basis of the skeleton of type I lamellarin alkaloids isolated from marine mollusks, tunicates and sponges [1,2] (Figure 1). Some of them have shown anticancer activity [3,4], including against multidrug-resistant cancer cell lines [5], as well as anti-HIV-1 activity at noncytotoxic concentration [6]. Synthetic chromeno[3,4-*b*]pyrrole derivatives **1** exhibited antibacterial activity against *Staphylococcus aureus* and *Escherichia coli* comparable to that of gentamicin [7]. Significant pharmacological activity was also found in chromeno[4,3-*b*]- and chromeno[3,4-*c*]pyrrole derivatives. For example, 1,3-diaryl-substituted pyrrolo[3,2-*c*]coumarin **2** is a benzodiazepine receptor ligand [8], while pyrrolo[3,4-*c*]coumarin-1-carboxylic acid **3** has shown high antibacterial activity against Gram-positive and Gram-negative bacteria [9] (Figure 1). Pericyclic pyrrolo[3,4-*c*]coumarin **4** emits blue light and may be tested as blue-light emitters with an electron-transporting ability [10] (Figure 1). In view of the above, the development of methods for the synthesis of these classes of compounds is being carried out by various scientific groups, and the search for new effective approaches to the design of chromenopyrrole scaffolds continues [11,12,13,14,15].

There are several strategies for the synthesis of the chromeno[3,4-*c*]pyrrole core. The most important ones are shown in Figure 1. Wang et al. developed an efficient method for the synthesis of pyrrolo[3,4-*c*]coumarins **5** via FeCl_3_-promoted three-component reaction between 2-hydroxy-β-nitrostyrenes, dimethyl acetylenedicarboxylate and amines [16]. Fused pyrroles **6** were obtained by Khavasi et al. [17] and Alizadeh et al. [18] from 3-acylcoumarins or salicylaldehydes and β-keto esters through a two- or three-component reaction using the Van Leusen protocol. Samanta et al. reported the synthesis of 2-(2,4-dihydrochromeno[3,4-*c*]pyrrol-1-yl)acetates **8** from 3-(3-formyl-2*H*-chromen-4-yl)acrylates **7** by treatment with hydroxylamine hydrochloride, amino alcohols or sodium azide [19,20]. Recently, a metal-free method for the synthesis of chromeno[3,4-*c*]pyrroles **9** based on nitrative cyclization of the corresponding 1,7-diynes with *tert*-butyl nitrite in the presence of water has been developed [21].

There are also reports on the synthesis of 1,3-diphenylpyrrole[3,4-*c*]coumarin by cycloaddition of the azomethine ylide generated from benzaldehyde and phenylglycine to 4-phenylsulfinylcoumarin [22], 2-benzyl-1-(pyridin-2-ylmethyl)-2,4-dihydrochromeno[3,4-*c*]pyrrole from 4-bromo-2*H*-chromene-3-carbaldehyde, benzylamine and 2-ethynylpyridine via the palladium-catalyzed one-pot Sonogashira reaction [23], and 2-benzyl-3-methyl-1-phenyl-2,4-dihydrochromeno[3,4-*c*]pyrrole from the corresponding 1,7-diyne through intramolecular [3+2] cycloaddition [24]. However, these methods are represented by single examples. Pyrrolocoumarin **4** was synthesized from 4-hydroxycoumarin, triethyl orthoformate and glycine in two steps in an overall yield of 81% [6]. A similar approach has been used to obtain 4,4-dialkyl-2-acetyl-2,4-dihydrochromeno[3,4-*c*]pyrroles from 2,2-dialkyl-4-oxochromane-3-carbaldehydes [25]. More recently, a copper-catalyzed [1,2]-Stevens-type asymmetric cascade cyclization/rearrangement of OTBS-substituted *N*-propargyl ynamides to form chromeno[3,4-*c*]pyrroles bearing a chiral C-4 stereocenter has been reported [26].

The Barton–Zard reaction is an efficient one-pot method for the synthesis of 5-unsubstituted pyrroles from readily available conjugated nitroalkenes and alkyl isocyanoacetates [27,28,29,30,31]. From this point of view, 3-nitro-2*H*-chromenes are suitable substrates for the design of a chromeno[3,4-*c*]pyrrole scaffold. Indeed, due to the presence of a β-nitrostyrene fragment in the molecule, 3-nitro-2*H*-chromenes are widely used in the synthesis of fused polyheterocyclic systems [32,33,34]. The introduction of a trifluoromethyl group into a drug molecule often leads to an increase in its physiological activity because of improvements to the transport characteristics of the drug and an increase in its metabolic stability [35,36]. We have recently developed methods for the synthesis of trifluoromethyl-substituted chromenopyrroli(zi)dines with pronounced cytotoxic activity against HeLa and RD cancer cells [37,38,39]. In this work, we report an eco-friendly and efficient approach to the synthesis of 2,4-dihydrochromeno[3,4-*c*]pyrroles **12** from 2-mono- and 2,2-disubsituted 3-nitrochromenes **10** and ethyl isocyanoacetate **11** via the Barton–Zard reaction (Figure 1).

## 2. Results and Discussion

In order to obtain chromeno[3,4-*c*]pyrroles **12**, we optimized conditions for the reaction of 2-trifluoromethyl-substituted chromene **10aa** with ethyl isocyanoacetate **11** to form chromeno[3,4-*c*]pyrrole **12aa** (Figure 2, Table 1).

The reaction was carried out at room temperature (method A) or at reflux (method B). Three bases (DBU, DABCO and K_2_CO_3_) and three solvents (THF, MeCN and EtOH) were tested. It was found that regardless of the base used, the starting chromene **10aa** was absent in the reaction mixture after 1 h or 0.5 h under the conditions of method A or B, respectively (monitoring by TLC). In ethanol, all three bases were effective both at room temperature and at boiling (entries 7–9). In MeCN, the yields of the product increased noticeably with a rise in the temperature (entries 4–6). In contrast, if THF was used as the solvent, the yields decreased at reflux when DBU or DABCO were used as bases (entries 1–2). The best yield of **12aa** (94%) was achieved when the reaction was carried out in ethanol at reflux using 1.5 equiv. K_2_CO_3_ (entry 10). A further increase in the amount of base did not significantly affect the yield of product (entries 11–12).

Next, under optimized conditions, the substrate scope for the synthesis of chromeno[3,4-*c*]pyrroles **12** were examined by varying the substituents R^1^–R^4^ in nitrochromenes **10** (Figure 3).

The substituents at the 2-position of chromene **10** had a notable effect on the yields of the products **12** (Figure 3). The highest yields (83–94%) were observed in the reactions of isonitrile **11** with 2-trifluoromethyl- and 2-phenyl-substituted chromenes **10aa**–**ag** and **10ba**–**bg**. The yields of 2-trichloromethyl-substituted pyrroles **12ca**–**cg** decreased by 7–14% compared to 2-trifluoromethyl-substituted analogs **12aa**–**ag**. Lower yields in the reactions involving chromenes **12ca**–**cg** were probably associated with the formation of 2-(dichloromethylidene)chromenes as a result of the elimination of HCl under the action of the base. We have already observed a similar process earlier in the reactions of these chromenes with sodium azide [40]. The introduction of a second substituent at the 2-position of chromenes **10aa**–**ag** or **10ba**–**bg** (Ph or CF_3_ group, respectively) reduced the yields of products **12da**–**dg** to 63–76% due to additional steric hindrances for attacking the double bond by the reagent. At the same time, the yields of pyrroles **12** were almost independent from the donor–acceptor properties of substituents R^3^ and R^4^. Replacement of the hydrogen atoms at the positions 6 and 8 of the starting chromenes **10** with the donor MeO or EtO groups only slightly reduced the yields of compounds **12**.

To test the scalability of the procedure for the synthesis of 4-substituted 2,4-dihydrochromeno[3,4-*c*]pyrroles **12**, the gram-scale reaction of chromene **10aa** (1.00 g) with isonitrile **11** (0.60 g) was carried out under the standard condition to obtain the target product **12aa** (1.20 g) in 94% yield (Figure 4).

The probable reaction mechanism includes the Michael addition of isonitrile **11** to chromene **10**, intramolecular cyclization of the nitronate anion **A** into pyrroline **B**, protonation of anion **B**, elimination of nitrous acid in pyrroline **C**, and 1,3-H shift in 3*H*-pyrrole **D** to form chromeno[3,4-*c*]pyrrole **12** (Figure 5) [27].

The ^1^H NMR spectra of chromeno[3,4-*c*]pyrroles **12** contained a slightly broadened singlet of the 2-NH proton in the range of 9.08–9.53 ppm. The H-3 proton manifested as a doublet, or as a doublet of doublets at 6.36–7.16 ppm, with a coupling constants ^3^*J*_H3,H2_ = 2.6–3.1 Hz and ^4^*J*_H3,H4_ = 0.7 Hz in the spectra of 2-monosubstituted chromenopyrroles **12aa**–**cg**, and as a doublet of quartets at 7.16–7.23 ppm with coupling constants ^3^*J*_H3,H2_ =2.6–2.8 Hz, ^5^*J*_H,F_ = 1.4–1.5 Hz in the spectra of 2,2-disubstituted chromenopyrroles **12da**–**dg**. The signal of the H-9 proton was unshielded with respect to the other protons of the benzene ring and resonated at 8.10–8.89 ppm. The ^19^F NMR spectra of products **12aa**–**ag** featured a doublet of the trifluoromethyl group at 82.6–83.0 ppm with ^3^*J*_F,H_ = 6.6–6.8 Hz, while in the spectra of compounds **12da**–**dg** this group manifested as a singlet in the range of 84.3–85.2 ppm. The ^13^C NMR spectra of these compounds contained quartets of the CF_3_ group and the C-4 atom in the range of 122.9–124.1 and 71.0–82.2 ppm with coupling constants 283.1–284.3 and 30.8–34.8 Hz, respectively.

To assess the possibility of using pyrroles **12** in organic synthesis, some transformations of the pyrrole ring were carried out (Figure 6). It was found that chromenopyrrole **12aa** was methylated at the nitrogen atom to form the *N*-methyl derivative **13** in 76% yield. Treatment of compound **12aa** with phenylboronic acid by the Chan–Evans–Lahm coupling reaction led to the corresponding *N*-phenylpyrrole **14** in 45% yield. 3-Bromo derivative **15** was obtained in 55% yield by bromination of compound **15** with *N*-bromosuccinimide.

In summary, a green and efficient regioselective method for the synthesis of 4-substituted 2,4-dihydrochromeno[3,4-*c*]pyrroles has been developed by the Barton–Zard reaction, using K_2_CO_3_ as a base and ethanol as a solvent. The availability of the starting 3-nitro-2*H*-chromenes, operational simplicity and scalability, as well as the possibility of further functionalization of the products, open up prospects for the synthesis of libraries of compounds bearing a chromeno[3,4-*c*]pyrrole framework, which are of undoubted interest for medicinal chemistry, especially due to the presence of the CF_3_ group.

## 3. Materials and Methods

### 3.1. General

IR spectra were recorded on a Shimadzu IRSpirit-T spectrometer using an attenuated total reflectance (ATR) unit (FTIR mode, ZnSe crystal); the absorbance maxima (*ν*) are reported in cm^–1^. NMR spectra (See Appendix A) were recorded on Bruker Avance III-500 (^1^H, 500 MHz; ^19^F, 471 MHz; ^13^C, 126 MHz) and Bruker DRX-400 (^1^H, 400 MHz; ^19^F, 376 MHz) spectrometers in CDCl_3_. The chemical shifts (*δ*) are reported in ppm relative to the internal standard TMS (^1^H NMR), C_6_F_6_ (^19^F NMR), and residual signal of the solvent (^13^C NMR). The HRMS spectra were obtained using the UHR-QqTOF maXis Impact HD (Bruker Daltonics, Billerica, MA, USA) mass spectrometer. Melting points were determined on an SMP40 apparatus. Monitoring of the reaction progress and assessment of the purity of synthesized compounds were carried out by TLC on Sorbfil PTSKh-AF-A-UF plates (eluent EtOAc–hexane, 1:3). All solvents used were dried and distilled by standard procedures. The starting chromenes **10** were prepared according to described procedures [41,42,43]. Compounds **13**–**15** were obtained according to the procedures analogous to those described in [44,45,46].

### 3.2. Synthesis of Compounds ***12aa**–**dg***

General procedure. To a mixture of the appropriate 3-nitro-2*H*-chromene **10** (0.5 mmol) and K_2_CO_3_ (104 mg, 0.75 mmol) in EtOH (4 mL), a solution of ethyl isocyanoacetate **11** (74 mg, 0.65 mmol) in EtOH (2 mL) was added dropwise with stirring. Then, the mixture was refluxed for 0.5 h with stirring (TLC control, EtOAc–hexane (1:3)). After completion of the reaction, 1 mL of 5% hydrochloric acid was added and the reaction mixture was evaporated under reduced pressure. Then, water (25 mL) was added to the residue, the precipitate was filtered, dried at 75 °C and recrystallized from a dichloromethane–hexane (2:1) system to give products **12** as beige powders.

*Ethyl 4-(trifluoromethyl)-2,4-dihydrochromeno[3,4-c]pyrrole-1-carboxylate* (**12aa**). Yield 146 mg (94%), mp 109–110 °C. IR (ATR) ν 3286 (NH), 1667 (C=O). ^1^H NMR (500 MHz, CDCl_3_) *δ* 1.43 (t, 3H, *J* = 7.1 Hz), 4.42 (q, 2H, *J* = 7.1 Hz), 5.57 (q, 1H, *J* = 6.6 Hz), 6.92 (d, 1H, *J* = 3.0 Hz), 7.04 (dd, 1H, *J* = 8.0, 1.4 Hz), 7.06 (td, 1H, *J* = 8.0, 1.4 Hz), 7.22 (td, 1H, *J* = 8.0, 1.5 Hz), 8.68 (dd, 1H, *J* = 8.0, 1.5 Hz), 9.29 (s, 1H); ^19^F NMR (471 MHz, CDCl_3_) *δ* 82.9 (d, *J* = 6.6 Hz, CF_3_); ^13^C NMR (126 MHz, CDCl_3_) *δ* 14.4, 61.0, 71.0 (q, ^2^*J*_CF_ = 34.2 Hz), 112.2, 116.9, 117.5, 117.8 (2C), 121.0, 122.5, 123.4 (q, ^1^*J*_CF_ = 283.8 Hz), 127.6, 129.2, 151.7, 160.1. HRMS (ESI) *m*/*z*: [M + H]^+^ calcd for C_15_H_13_F_3_NO_3_ 312.0843, found 312.0846.

*Ethyl 8-methoxy-4-(trifluoromethyl)-2,4-dihydrochromeno[3,4-c]pyrrole-1-carboxylate* (**12ab**). Yield 145 mg (85%), mp 120–123 °C. IR (ATR) ν 3275 (NH), 1676 (C=O). ^1^H NMR (400 MHz, CDCl_3_) *δ* 1.42 (t, 3H, *J* = 7.1 Hz), 4.42 (q, 2H, *J* = 7.1 Hz), 5.52 (qd, 1H, *J* = 6.7, 0.4 Hz), 6.78 (dd, 1H, *J* = 8.8, 3.1 Hz), 6.91 (d, 1H, *J* = 3.1 Hz), 6.96 (d, 1H, *J* = 8.8 Hz), 8.37 (d, 1H, *J* = 3.1 Hz), 9.29 (s, 1H); ^19^F NMR (376 MHz, CDCl_3_) *δ* 83.0 (d, *J* = 6.7 Hz, CF_3_); ^13^C NMR (126 MHz, CDCl_3_) *δ* 14.5, 55.7, 61.0, 70.9 (q, ^2^*J*_CF_ = 33.9 Hz), 112.1, 112.5, 115.4, 117.4, 117.5, 117.7, 118.3, 121.4, 123.4 (d, ^1^*J*_CF_ = 283.6 Hz), 145.7, 154.8, 160.0. HRMS (ESI) *m*/*z*: [M + H]^+^ calcd for C_16_H_15_F_3_NO_4_ 342.0948, found 342.0953.

*Ethyl 6-ethoxy-4-(trifluoromethyl)-2,4-dihydrochromeno[3,4-c]pyrrole-1-carboxylate* (**12ac**). Yield 147 mg (83%), mp 115–117 °C. IR (ATR) ν 3393, 1698, 1567, 1470, 1455, 1414, 1351, 1311. ^1^H NMR (400 MHz, CDCl_3_) *δ* 1.42 (t, 3H, *J* = 7.1 Hz), 1.44 (t, 3H, *J* = 7.0 Hz), 4.13 (q, 2H, *J* = 7.0 Hz), 4.41 (q, 2H, *J* = 7.1 Hz), 5.66 (q, 1H, *J* = 6.8 Hz), 6.90 (dd, 1H, *J* = 8.0, 1.4 Hz), 6.93 (d, 1H, *J* = 3.1 Hz), 6.98 (t, 1H, *J* = 8.0 Hz), 8.30 (dd, 1H, *J* = 8.0, 1.5 Hz), 9.33 (s, 1H); ^19^F NMR (376 MHz, CDCl_3_) *δ* 82.9 (d, *J* = 6.8 Hz); ^13^C NMR (126 MHz, CDCl_3_) *δ* 14.4, 14.9, 61.0, 65.4, 70.9 (q, ^2^*J*_CF_ = 34.3 Hz), 112.3, 115.0, 117.6, 117.9, 118.8, 120.1, 121.0, 121.9, 123.4 (q, ^1^*J*_CF_ = 284.3 Hz), 141.8, 147.8, 160.1. HRMS (ESI) *m*/*z*: [M + H]^+^ calcd for C_17_H_17_F_3_NO_4_ 356.1105, found 356.1111.

*Ethyl 8-chloro-4-(trifluoromethyl)-2,4-dihydrochromeno[3,4-c]pyrrole-1-carboxylate* (**12ad**). Yield 154 mg (89%), mp 169–170 °C. IR (ATR) ν 3279 (NH), 1667 (C=O). ^1^H NMR (400 MHz, CDCl_3_) *δ* 1.47 (t, 3H, *J* = 7.1 Hz), 4.44 (q, 2H, *J* = 7.1 Hz), 5.57 (q, 1H, *J* = 6.6 Hz), 6.94 (d, 1H, *J* = 3.1 Hz), 6.97 (d, 1H, *J* = 8.7 Hz), 7.16 (dd, 1H, *J* = 8.7, 2.6 Hz), 8.70 (d, 1H, *J* = 2.6 Hz), 9.47 (s, 1H); ^19^F NMR (376 MHz, CDCl_3_) *δ* 82.8 (d, *J* = 6.6 Hz); ^13^C NMR (126 MHz, CDCl_3_) *δ* 14.3, 61.4, 71.0 (q, ^2^*J*_CF_ = 34.5 Hz), 112.0, 117.9, 118.1 (2C), 119.1, 119.5, 123.2 (q, ^1^*J*_CF_ = 283.9 Hz), 127.3, 127.4, 128.8, 150.2, 160.2. HRMS (ESI) *m*/*z*: [M + H]^+^ calcd for C_15_H_12_ClF_3_NO_3_ 346.0454, found 346.0455.

*Ethyl 8-bromo-4-(trifluoromethyl)-2,4-dihydrochromeno[3,4-c]pyrrole-1-carboxylate* (**12ae**). Yield 176 mg (90%), mp 177–178 °C. IR (ATR) ν 3281 (NH), 1666 (C=O). ^1^H NMR (400 MHz, CDCl_3_) *δ* 1.49 (t, 3H, *J* = 7.1 Hz), 4.45 (q, 2H, *J* = 7.1 Hz), 5.57 (q, 1H, *J* = 6.6 Hz), 6.92 (d, 1H, *J* = 8.7 Hz), 6.94 (d, 1H, *J* = 3.1 Hz), 7.30 (dd, 1H, *J* = 8.6, 2.4 Hz), 8.84 (d, 1H, *J* = 2.5 Hz), 9.39 (s, 1H); ^19^F NMR (376 MHz, CDCl_3_) *δ* 82.8 (d, *J* = 6.6 Hz); ^13^C NMR (126 MHz, CDCl_3_) *δ* 14.4, 61.5, 71.0 (q, ^2^*J*_CF_ = 34.0 Hz), 112.0, 114.9, 117.9, 118.1, 118.6, 119.3, 119.6, 123.2 (q, ^1^*J*_CF_ = 283.7 Hz), 130.2, 131.7, 150.7, 160.2. HRMS (ESI) *m*/*z*: [M + H]^+^ calcd for C_15_H_12_BrF_3_NO_3_ 389.9947, found 389.9948.

*Ethyl 6,8-dichloro-4-(trifluoromethyl)-2,4-dihydrochromeno[3,4-c]pyrrole-1-carboxylate* (**12af**). Yield 167 mg (88%), mp 187–190 °C. IR (ATR): ν 3278 (NH), 1671 (C=O). ^1^H NMR (400 MHz, CDCl_3_) *δ* 1.47 (t, 3H, *J* = 7.1 Hz), 4.43 (dq, 1H, *J* = 10.3, 7.1 Hz), 4.47 (dq, 1H, *J* = 10.3, 7.1 Hz), 5.68 (q, 1H, *J* = 6.6 Hz), 6.97 (d, 1H, *J* = 3.1 Hz), 7.29 (d, 1H, *J* = 2.5 Hz), 8.68 (d, 1H, *J* = 2.5 Hz), 9.44 (s, 1H); ^19^F NMR (376 MHz, CDCl_3_) *δ* 82.6 (d, *J* = 6.6 Hz); ^13^C NMR (126 MHz, CDCl_3_) *δ* 14.4, 61.5, 71.4 (q, ^2^*J*_CF_ = 34.8 Hz), 111.9, 118.1, 118.3, 119.0, 120.3, 122.8, 122.9 (q, ^1^*J*_CF_ = 283.6 Hz), 125.9, 127.3, 129.1, 146.1, 159.9. HRMS (ESI) *m*/*z*: [M + H]^+^ calcd for C_15_H_11_Cl_2_F_3_NO_3_ 380.0063, found 380.0059.

*Ethyl 6,8-dibromo-4-(trifluoromethyl)-2,4-dihydrochromeno[3,4-c]pyrrole-1-carboxylate* (**12ag**). Yield 213 mg (91%), mp 187–188 °C. IR (ATR): ν 3278 (NH), 1668 (C=O). ^1^H NMR (400 MHz, CDCl_3_) *δ* 1.47 (t, 3H, *J* = 7.1 Hz), 4.42 (dq, 1H, *J* = 11.0, 7.1 Hz), 4.46 (dq, 1H, *J* = 11.0, 7.1 Hz), 5.68 (q, 1H, *J* = 6.5 Hz), 6.97 (d, 1H, *J* = 3.1 Hz), 7.58 (d, 1H, *J* = 2.3 Hz), 8.84 (d, 1H, *J* = 2.3 Hz), 9.46 (s, 1H); ^19^F NMR (376 MHz, CDCl_3_) *δ* 82.6 (d, *J* = 6.6 Hz); ^13^C NMR (126 MHz, CDCl_3_) *δ* 14.4, 61.6, 71.4 (q, ^2^*J*_CF_ = 34.7 Hz), 111.8, 112.0, 114.8, 118.1, 118.3, 118.8, 120.8, 122.9 (d, ^1^*J*_CF_ = 284.1 Hz), 129.4, 134.6, 147.5, 159.9. HRMS (ESI) *m*/*z*: [M + H]^+^ calcd for C_15_H_11_Br_2_F_3_NO_3_ 467.9051, found 467.9051.

*Ethyl 4-phenyl-2,4-dihydrochromeno[3,4-c]pyrrole-1-carboxylate* (**12ba**). Yield 136 mg (85%), mp 91–93 °C. IR (ATR) ν 3298 (NH), 1660 (C=O). ^1^H NMR (400 MHz, CDCl_3_) *δ* 1.40 (t, 3H, *J* = 7.1 Hz,), 4.38 (dq, 1H, *J* = 10.9, 7.1 Hz), 4.44 (dq, 1H, *J* = 10.9, 7.1 Hz), 6.05 (s, 1H), 6.36 (d, 1H, *J* = 2.9 Hz), 7.01 (dd, 1H, *J* = 8.0, 1.1 Hz), 7.04 (td, 1H, *J* = 8.0, 1.1 Hz), 7.18 (td, 1H, *J* = 8.0, 1.5 Hz), 7.33–7.41 (m, 3H), 7.44–7.49 (m, 2H), 8.59 (dd, 1H, *J* = 8.0, 1.5 Hz), 9.18 (s, 1H); ^13^C NMR (126 MHz, CDCl_3_) *δ* 14.5, 60.7, 76.0, 117.1, 117.6, 119.8, 121.5, 121.8, 123.4, 125.6, 127.6 (2C), 127.6, 128.5 (2C), 128.5, 128.8, 139.9, 153.9, 160.7. HRMS (ESI) *m*/*z*: [M + H]^+^ calcd for C_20_H_18_NO_3_ 320.1281, found 320.1286.

*Ethyl 8-methoxy-4-phenyl-2,4-dihydrochromeno[3,4-c]pyrrole-1-carboxylate* (**12bb**). Yield 145 mg (83%), mp 140–142 °C. IR (ATR) ν 3281 (NH), 1673 (C=O). ^1^H NMR (400 MHz, CDCl_3_) *δ* 1.42 (t, 3H, *J* = 7.1 Hz), 3.86 (s, 3H), 4.40 (dq, 1H, *J* = 10.8, 7.1 Hz), 4.43 (dq, 1H, *J* = 10.8, 7.1 Hz), 6.01 (s, 1H), 6.38 (dd, 1H, *J* = 2.9, 0.7 Hz), 6.77 (dd, 1H, *J* = 8.8, 3.1 Hz), 6.94 (d, 1H, *J* = 8.8 Hz), 7.34–7.42 (m, 3H), 7.45–7.50 (m, 2H), 8.28 (d, 1H, *J* = 3.1 Hz), 9.08 (s, 1H); ^13^C NMR (126 MHz, CDCl_3_) *δ* 14.6, 55.8, 60.7, 75.9, 112.1, 115.0, 117.0, 117.1, 118.0, 120.3, 121.9, 123.6, 127.6, 128.4 (2C), 128.5 (2C), 139.9, 147.9, 154.4, 160.5. HRMS (ESI) *m*/*z*: [M + H]^+^ calcd for C_21_H_20_NO_4_ 350.1387, found 350.1392.

*Ethyl 6-ethoxy-4-phenyl-2,4-dihydrochromeno[3,4-c]pyrrole-1-carboxylate* (**12bc**). Yield 156 mg (86%), mp 85–87 °C. IR (ATR) ν 3323 (NH), 1701 (C=O). ^1^H NMR (400 MHz, CDCl_3_) *δ* 1.37 (t, 3H, *J* = 7.0 Hz), 1.40 (t, 3H, *J* = 7.1 Hz), 4.06 (dq, 1H, *J* = 9.8, 7.0 Hz), 4.09 (dq, 1H, *J* = 9.8, 7.0 Hz), 4.37 (dq, 1H, *J* = 11.0, 7.1 Hz), 4.40 (dq, 1H, *J* = 11.0, 7.1 Hz), 6.16 (s, 1H), 6.49 (dd, 1H, *J* = 2.8 Hz), 6.85 (dd, 1H, *J* = 8.0, 1.5 Hz), 6.95 (t, 1H, *J* = 8.0 Hz), 7.28–7.36 (m, 3H), 7.43–7.48 (m, 2H), 8.20 (dd, 1H, *J* = 8.0, 1.5 Hz), 9.19 (s, 1H); ^13^C NMR (126 MHz, CDCl_3_) δ 14.4, 14.9, 60.7, 65.1, 75.4, 114.5, 117.0, 117.2, 120.2, 120.9, 121.1, 121.4, 123.0, 127.3 (2C), 128.1, 128.3 (2C), 140.0, 143.7, 148.3, 160.6. HRMS (ESI) *m*/*z*: [M + H]^+^ calcd for C_22_H_22_NO_4_ 364.1543, found 364.1550.

*Ethyl 8-chloro-4-phenyl-2,4-dihydrochromeno[3,4-c]pyrrole-1-carboxylate* (**12bd**). Yield 154 mg (87%), mp 145–146 °C. IR (ATR) ν 3303 (NH), 1668 (C=O). ^1^H NMR (500 MHz, CDCl_3_) *δ* 1.47 (t, 3H, *J* = 7.1 Hz), 4.41 (dq, 1H, *J* = 10.9, 7.1 Hz), 4.44 (dq, 1H, *J* = 10.9, 7.1 Hz), 6.06 (s, 1H), 6.42 (d, 1H, *J* = 2.6 Hz), 6.93 (d, 1H, *J* = 8.6 Hz), 7.12 (dd, 1H, *J* = 8.6, 2.6 Hz), 7.35–7.41 (m, 3H), 7.43–7.46 (m, 2H), 8.60 (d, 1H, *J* = 2.6 Hz), 9.20 (s, 1H); ^13^C NMR (126 MHz, CDCl_3_) *δ* 14.4, 61.2, 76.1, 117.3, 117.5, 118.8, 120.1, 121.1, 123.1, 126.7, 127.3, 127.6 (2C), 128.4, 128.6 (2C), 128.7, 139.5, 152.4, 160.7. HRMS (ESI) *m*/*z*: [M + H]^+^ calcd for C_20_H_17_ClNO_3_ 354.0891 found 354.0895.

*Ethyl 8-bromo-4-phenyl-2,4-dihydrochromeno[3,4-c]pyrrole-1-carboxylate* (**12be**). Yield 177 mg (89%), mp 141–143 °C. IR (ATR) ν 3298 (NH), 1665 (C=O). ^1^H NMR (400 MHz, CDCl_3_) *δ* 1.48 (t, 3H, *J* = 7.1 Hz), 4.41 (dq, 1H, *J* = 10.8, 7.1 Hz), 4.47 (dq, 1H, *J* = 10.8, 7.1 Hz), 6.06 (s, 1H), 6.42 (d, 1H, *J* = 2.9 Hz), 6.88 (d, 1H, *J* = 8.6 Hz), 7.31–7.42 (m, 3H), 7.42–7.48 (m, 2H), 8.73 (d, 1H, *J* = 2.5 Hz), 9.21 (s, 1H); ^13^C NMR (126 MHz, CDCl_3_) *δ* 14.4, 61.2, 76.1, 114.2, 117.3, 117.5, 119.3, 119.8, 121.7, 123.0, 127.6 (2C), 128.6 (2C), 128.7, 130.1, 131.3, 139.5, 152.9, 160.7. HRMS (ESI) *m/z*: [M + H]^+^ calcd for C_20_H_17_BrNO_3_ 398.0386, found 398.0394.

*Ethyl 6,8-dichloro-4-phenyl-2,4-dihydrochromeno[3,4-c]pyrrole-1-carboxylate* (**12bf**). Yield 163 mg (84%), mp 138–140 °C. IR (ATR) ν 3317 (NH), 1662 (C=O). ^1^H NMR (500 MHz, CDCl_3_) *δ* 1.46 (t, 3H, *J* = 7.1 Hz), 4.41 (dq, 1H, *J* = 10.8, 7.1 Hz), 4.44 (dq, 1H, *J* = 10.8, 7.1 Hz), 6.22 (s, 1H), 6.57 (d, 1H, *J* = 2.9 Hz), 7.24 (d, 1H, *J* = 2.5 Hz), 7.34–7.39 (m, 3H), 7.41–7.45 (m, 2H), 8.53 (d, 1H, *J* = 2.5 Hz), 9.30 (s, 1H); ^13^C NMR (126 MHz, CDCl_3_) *δ* 14.4, 61.3, 76.1, 117.3, 117.8, 119.4, 122.3, 122.5, 123.4, 125.8, 126.3, 127.3 (2C), 128.5 (2C), 128.6, 128.6, 139.0, 148.1, 160.5. HRMS (ESI) *m*/*z*: [M + H]^+^ calcd for C_20_H_16_Cl_2_NO_3_ 388.0502, found 388.0505.

*Ethyl 6,8-dibromo-4-phenyl-2,4-dihydrochromeno[3,4-c]pyrrole-1-carboxylate* (**12bg**). Yield 205 mg (86%), mp 185–187 °C. IR (ATR) ν 3265 (NH), 1667 (C=O). ^1^H NMR (400 MHz, CDCl_3_) *δ* 1.47 (t, 3H, *J* = 7.1 Hz), 4.42 (dq, 1H, *J* = 10.8, 7.1 Hz), 4.44 (dq, 1H, *J* = 10.8, 7.1 Hz), 6.24 (s, 1H), 6.58 (d, 1H, *J* = 2.9 Hz), 7.30–7.40 (m, 3H), 7.40–7.46 (m, 2H), 7.53 (d, 1H, *J* = 2.5 Hz), 8.70 (d, 1H, *J* = 2.5 Hz), 9.26 (s, 1H); ^13^C NMR (126 MHz, CDCl_3_) *δ* 14.4, 61.3, 76.0, 112.6, 113.9, 117.2, 117.9, 119.2, 122.4, 122.7, 127.2 (2C), 128.5 (2C), 128.6, 129.3, 134.0, 139.0, 149.5, 160.4. HRMS (ESI) *m*/*z*: [M + H]^+^ calcd for C_20_H_16_Br_2_NO_3_ 475.9491, found 475.9496.

*Ethyl 4-(trichloromethyl)-2,4-dihydrochromeno[3,4-c]pyrrole-1-carboxylate* (**12ca**). Yield 144 mg (80%), mp 135–137 °C. IR (ATR) ν 3278 (NH), 1644 (C=O). ^1^H NMR (400 MHz, CDCl_3_) *δ* 1.43 (t, 3H, *J* = 7.1 Hz), 4.43 (q, 2H, *J* = 7.1 Hz), 5.76 (s, 1H), 7.04 (dd, 1H, *J* = 8.0, 1.3 Hz), 7.06 (td, 1H, *J* = 8.0, 1.3 Hz), 7.12 (d, 1H, *J* = 3.1 Hz), 7.23 (td, 1H, *J* = 8.0, 1.6 Hz), 8.72 (dd, 1H, *J* = 8.0, 1.6 Hz), 9.37 (s, 1H); ^13^C NMR (126 MHz, CDCl_3_) *δ* 14.5, 61.0, 81.9, 101.5, 113.8, 116.8, 116.9, 117.9, 120.4, 121.7, 122.1, 127.4, 129.3, 152.0, 160.2. HRMS (ESI) *m*/*z*: [M + H]^+^ calcd for C_15_H_13_Cl_3_NO_3_ 359.9956, found 359.9960.

*Ethyl 8-methoxy-4-(trichloromethyl)-2,4-dihydrochromeno[3,4-c]pyrrole-1-carboxylate* (**12cb**). Yield 152 mg (78%), mp 155–157 °C. IR (ATR) ν 3301 (NH), 1683 (C=O). ^1^H NMR (400 MHz, CDCl_3_) *δ* 1.43 (t, 3H, *J* = 7.1 Hz), 3.85 (s, 3H), 4.43 (q, 2H, *J* = 7.1 Hz), 5.70 (s, 1H), 6.79 (dd, 1H, *J* = 8.8, 3.1 Hz), 6.99 (d, 1H, *J* = 8.8 Hz), 7.11 (d, 1H, *J* = 3.1 Hz), 8.41 (d, 1H, *J* = 3.1 Hz), 9.37 (s, 1H); ^13^C NMR (126 MHz, CDCl_3_) *δ* 14.5, 55.7, 60.9, 81.9, 101.5, 111.8, 114.0, 115.5, 116.9, 117.3, 118.3, 120.4, 122.1, 146.0, 154.5, 160.1. HRMS (ESI) *m/z*: [M + H]^+^ calcd for C_16_H_15_Cl_3_NO_4_ 390.0062, found 390.0067.

*Ethyl 6-ethoxy-4-(trichloromethyl)-2,4-dihydrochromeno[3,4-c]pyrrole-1-carboxylate* (**12cc**). Yield 152 mg (75%), mp 135–137 °C. IR (ATR) ν 3305 (NH), 1682 (C=O). ^1^H NMR (400 MHz, CDCl_3_) *δ* 1.42 (t, 3H, *J* = 7.1 Hz), 1.46 (t, 3H, *J* = 7.0 Hz), 4.11 (dq, 1H, *J* = 9.7, 7.1 Hz), 4.18 (dq, 1H, *J* = 9.7, 7.1 Hz), 4.42 (q, 2H, *J* = 7.1 Hz), 5.86 (s, 1H), 6.90 (dd, 1H, *J* = 8.0, 1.6 Hz), 6.97 (t, 1H, *J* = 8.0 Hz), 7.13 (d, 1H, *J* = 3.1 Hz), 8.34 (dd, 1H, *J* = 8.0, 1.6 Hz), 9.38 (s, 1H); ^13^C NMR (126 MHz, CDCl_3_) *δ* 14.4, 15.0, 60.9, 65.4, 81.9, 101.4, 113.8, 114.8, 117.0, 118.9, 119.7, 120.4, 121.5, 121.7, 142.1, 147.7, 160.2. HRMS (ESI) *m*/*z*: [M + H]^+^ calcd for C_17_H_17_Cl_3_NO_4_ 404.021, found 404.0219.

*Ethyl 8-chloro-4-(trichloromethyl)-2,4-dihydrochromeno[3,4-c]pyrrole-1-carboxylate* (**12cd**) Yield 156 mg (79%), mp 195–197 °C. IR (ATR) ν 3278 (NH), 1664 (C=O). ^1^H NMR (400 MHz, CDCl_3_) *δ* 1.48 (t, 3H, *J* = 7.1 Hz), 4.40 (dq, 1H, *J* = 10.8, 7.1 Hz), 4.47 (dq, 1H, *J* = 10.8, 7.1 Hz), 5.75 (s, 1H), 7.00 (d, 1H, *J* = 8.6 Hz), 7.14 (d, 1H, *J* = 3.0 Hz), 7.17 (dd, 1H, *J* = 8.6, 2.6 Hz), 8.75 (d, 1H, *J* = 2.6 Hz), 9.44 (s, 1H); ^13^C NMR (126 MHz, CDCl_3_) *δ* 14.4, 61.4, 82.0, 101.3, 113.7, 117.3, 118.0, 119.2, 120.3, 120.5, 127.1, 127.1, 128.9, 150.5, 160.2. HRMS (ESI) *m*/*z*: [M + H]^+^ calcd for C_15_H_12_Cl_4_NO_3_ 393.9566, found 393.9569.

*Ethyl 8-bromo-4-(trichloromethyl)-2,4-dihydrochromeno[3,4-c]pyrrole-1-carboxylate* (**12ce**). Yield 176 mg (80%), mp 180–181 °C. IR (ATR) ν 3277 (NH), 1664 (C=O). ^1^H NMR (400 MHz, CDCl_3_) *δ* 1.49 (t, 3H, *J* = 7.1 Hz), 4.44 (dq, 1H, *J* = 10.8, 7.1 Hz), 4.47 (dq, 1H, *J* = 10.8, 7.1 Hz), 5.75 (s, 1H), 6.95 (d, 1H, *J* = 8.6 Hz), 7.14 (d, 1H, *J* = 3.1 Hz), 7.31 (dd, 1H, *J* = 8.6, 2.4 Hz), 8.87 (d, 1H, *J* = 2.4 Hz), 9.47 (s, 1H); ^13^C NMR (126 MHz, CDCl_3_) *δ* 14.4, 61.5, 81.9, 101.3, 113.6, 114.5, 117.3, 118.5, 119.8, 120.0, 120.6, 129.9, 131.8, 151.0, 160.3. HRMS (ESI) *m*/*z*: [M + H]^+^ calcd for C_15_H_12_BrCl_3_NO_3_ 437.9061, found 437.9067.

*Ethyl 6,8-dichloro-4-(trichloromethyl)-2,4-dihydrochromeno[3,4-c]pyrrole-1-carboxylate* (**12cf**). Yield 174 mg (81%), mp 143–144 °C. IR (ATR) ν 3282 (NH), 1668 (C=O). ^1^H NMR (400 MHz, CDCl_3_) *δ* 1.47 (t, 3H, *J* = 7.1 Hz), 4.43 (dq, 1H, *J* = 10.8, 7.1 Hz), 4.46 (dq, 1H, *J* = 10.8, 7.1 Hz), 5.86 (s, 1H), 7.16 (d, 1H, *J* = 3.0 Hz), 7.29 (d, 1H, *J* = 2.5 Hz), 8.71 (d, 1H, *J* = 2.5 Hz), 9.53 (s, 1H); ^13^C NMR (126 MHz, CDCl_3_) *δ* 14.4, 61.5, 82.3, 100.8, 113.7, 117.6, 119.7, 120.5, 120.6, 122.7, 125.6, 126.9, 129.1, 146.5, 160.0. HRMS (ESI) *m*/*z*: [M + H]^+^ calcd for C_15_H_11_Cl_5_NO_3_ 427.9176, found 427.9174.

*Ethyl 6,8-dibromo-4-(trichloromethyl)-2,4-dihydrochromeno[3,4-c]pyrrole-1-carboxylate* (**12cg**). Yield 207 mg (80%), mp 175–178 °C. IR (ATR) ν 3284 (NH), 1664 (C=O). ^1^H NMR (400 MHz, CDCl_3_) *δ* 1.48 (t, 3H, *J* = 7.1 Hz), 4.43 (dq, 1H, *J* = 10.8, 7.1 Hz), 4.47 (dq, 1H, *J* = 10.8, 7.1 Hz), 5.87 (s, 1H), 7.16 (d, 1H, *J* = 3.1 Hz), 7.59 (d, 1H, *J* = 2.3 Hz), 8.89 (d, 1H, *J* = 2.3 Hz), 9.52 (s, 1H); ^13^C NMR (126 MHz, CDCl_3_) *δ* 14.4, 61.5, 82.5, 100.7, 111.6, 113.7, 114.5, 117.6, 119.6, 120.5, 120.9, 129.1, 134.6, 148.0, 160.0. HRMS (ESI) *m/z*: [M + H]^+^ calcd for C_15_H_11_Br_2_Cl_3_NO_3_ 515.8166, found 515.8174.

*Ethyl 4-phenyl-4-(trifluoromethyl)-2,4-dihydrochromeno[3,4-c]pyrrole-1-carboxylate* (**12da**). Yield 141 mg (73%), mp 100–102 °C. IR (ATR) ν 3366 (NH), 1707 (C=O). ^1^H NMR (400 MHz, CDCl_3_) *δ* 1.42 (t, 3H, *J* = 7.1 Hz), 4.40 (dq, 1H, *J* = 10.8, 7.1 Hz), 4.43 (dq, 1H, *J* = 10.8, 7.1 Hz), 7.00 (ddd, 1H, *J* = 8.0, 7.6, 1.9 Hz), 7.14–7.22 (m, 3H), 7.24–7.31 (m, 3H), 7.48–7.54 (m, 2H), 8.53 (dd, 1H, *J* = 7.9, 1.2 Hz), 9.37 (s, 1H); ^19^F NMR (376 MHz, CDCl_3_) *δ* 85.2 (s, CF_3_); ^13^C NMR (126 MHz, CDCl_3_) *δ* 14.4, 61.0, 80.7 (q, *J* = 30.8 Hz), 117.0, 117.5, 117.9, 118.1, 118.7, 121.0, 122.5, 124.1 (q, ^1^*J*_CF_ = 283.7 Hz), 127.7, 128.0 (2C), 128.1 (2C), 129.0, 129.2, 135.4, 150.7, 160.2. HRMS (ESI) *m*/*z*: [M + H]^+^ calcd for C_21_H_17_F_3_NO_3_ 388.1155, found 388.1153.

*Ethyl 8-methoxy-4-phenyl-4-(trifluoromethyl)-2,4-dihydrochromeno[3,4-c]pyrrole-1-carboxylate* (**12db**). Yield 136 mg (65%), mp 105–107 °C. IR (ATR) ν 3346 (NH), 1714 (C=O). ^1^H NMR (400 MHz, CDCl_3_) *δ* 1.41 (t, 3H, *J* = 7.1 Hz), 3.78 (s, 3H), 4.39 (dq, 1H, *J* = 10.9, 7.1 Hz), 4.42 (dq, 1H, *J* = 10.9, 7.1 Hz), 6.76 (dd, 1H, *J* = 8.8, 3.1 Hz), 7.08 (d, 1H, *J* = 8.8 Hz), 7.16 (dq, 1H, *J* = 2.6, 1.4 Hz), 7.24–7.30 (m, 3H), 7.46–7.52 (m, 2H), 8.21 (d, 1H, *J* = 3.1 Hz), 9.45 (s, 1H); ^19^F NMR (376 MHz, CDCl_3_) *δ* 84.3 (s, CF_3_); ^13^C NMR (126 MHz, CDCl_3_) *δ* 14.5, 55.6, 61.0, 80.6 (q, ^2^*J*_CF_ = 30.8 Hz), 112.1, 115.4, 117.2, 117.5, 117.8, 118.7, 119.2, 121.3, 124.1 (q, ^1^*J*_CF_ = 283.6 Hz), 127.9 (2C), 128.2 (2C), 129.0, 135.3, 144.6, 154.8, 160.1. HRMS (ESI) *m*/*z*: [M + H]^+^ calcd for C_22_H_19_F_3_NO_4_ 418.1261, found 418.1259.

*Ethyl 6-ethoxy-4-phenyl-4-(trifluoromethyl)-2,4-dihydrochromeno[3,4-c]pyrrole-1-carboxylate* (**12dc**). Yield 136 mg (63%), mp 137–138 °C. IR (ATR) ν 3291 (NH), 1668 (C=O). ^1^H NMR (400 MHz, CDCl_3_) *δ* 1.41 (t, 3H, *J* = 7.1 Hz), 1.53 (t, 3H, *J* = 7.0 Hz), 4.13 (dq, 1H, *J* = 9.5, 7.0 Hz), 4.23 (dq, 1H, *J* = 9.5, 7.0 Hz), 4.38 (dq, 1H, *J* = 11.0, 7.1 Hz), 4.42 (dq, 1H, *J* = 11.0, 7.1 Hz), 6.83 (dd, 1H, *J* = 8.0, 1.4 Hz), 6.91 (t, 1H, *J* = 8.0 Hz), 7.21 (dq, 1H, *J* = 2.8, 1.4 Hz), 7.23–7.28 (m, 3H), 7.56–7.63 (m, 2H), 8.10 (dd, 1H, *J* = 8.0, 1.4 Hz), 9.39 (s, 1H); ^19^F NMR (376 MHz, CDCl_3_) *δ* 84.3 (s, CF_3_); ^13^C NMR (126 MHz, CDCl_3_) *δ* 14.4, 15.0, 60.9, 65.0, 81.2 (q, ^2^*J*_CF_ = 31.1 Hz), 114.3, 117.5, 117.6, 117.7, 119.8, 120.1, 121.1, 122.2, 124.1 (q, ^1^*J*_CF_ = 283.1 Hz), 127.9 (2C), 128.0 (2C), 129.0, 135.0, 140.6, 148.5, 160.2. HRMS (ESI) *m*/*z*: [M + H]^+^ calcd for C_23_H_21_F_3_NO_4_ 432.1417, found 432.1421.

*Ethyl 8-chloro-4-phenyl-4-(trifluoromethyl)-2,4-dihydrochromeno[3,4-c]pyrrole-1-carboxylate* (**12dd**). Yield 160 mg (76%), mp 158–160 °C. IR (ATR) ν 3457 (NH), 1720 (C=O). ^1^H NMR (400 MHz, CDCl_3_) *δ* 1.47 (t, 3H, *J* = 7.1 Hz), 4.41 (dq, 1H, *J* = 10.9, 7.1 Hz), 4.46 (dq, 1H, *J* = 10.9, 7.1 Hz), 7.05 (d, 1H, *J* = 8.6 Hz), 7.19 (dq, 1H, *J* = 2.6, 1.4 Hz), 7.27–7.31 (m, 4H), 7.43–7.49 (m, 2H), 8.68 (d, 1H, *J* = 2.4 Hz), 9.50 (s, 1H); ^19^F NMR (376 MHz, CDCl_3_) *δ* 84.3. (s, CF_3_); ^13^C NMR (126 MHz, CDCl_3_) *δ* 14.4, 61.5, 80.9 (q, ^2^*J*_CF_ = 31.2 Hz), 115.1, 116.9, 118.0, 118.1, 119.3, 119.8, 120.6, 123.9 (q, ^1^*J*_CF_ = 283.6 Hz), 128.1 (4C), 129.3, 130.2, 131.7, 134.9, 149.8, 160.3. HRMS (ESI) *m/z*: [M + H]^+^ calcd for C_21_H_16_ClF_3_NO_3_ 411.0765, found 422.0760.

*Ethyl 8-bromo-4-phenyl-4-(trifluoromethyl)-2,4-dihydrochromeno[3,4-c]pyrrole-1-carboxylate* (**12de**). Yield 168 mg (72%), mp 153–155 °C. IR (ATR) ν 3283 (NH), 1670 (C=O). ^1^H NMR (400 MHz, CDCl_3_) *δ* 1.47 (t, 3H, *J* = 7.1 Hz), 4.41 (dq, 1H, *J* = 10.9, 7.1 Hz), 4.46 (dq, 1H, *J* = 10.9, 7.1 Hz), 7.05 (d, 1H, *J* = 8.6 Hz), 7.19 (dq, 1H, *J* = 2.8, 1.5 Hz), 7.27–7.31 (m, 4H), 7.43–7.50 (m, 2H), 8.68 (d, 1H, *J* = 2.4 Hz), 9.47 (s, 1H); ^19^F NMR (376 MHz, CDCl_3_) *δ* 84.3 (s, CF_3_); ^13^C NMR (126 MHz, CDCl_3_) *δ* 14.4, 61.5, 80.9 (q, ^2^*J*_CF_ = 31.2 Hz), 115.1, 116.9, 118.0, 118.1, 119.3, 119.8, 120.6, 123.9 (q, *J* = 283.6 Hz), 128.1 (4C), 129.2, 130.2, 131.7, 134.9, 149.8, 160.3. HRMS (ESI) *m*/*z*: [M + H]^+^ calcd for C_21_H_16_BrF_3_NO_3_ 466.0260, found 466.0262.

*Ethyl 6,8-dichloro-4-phenyl-4-(trifluoromethyl)-2,4-dihydrochromeno[3,4-c]pyrrole-1-carboxylate* (**12df**). Yield 155 mg (68%), mp 172–173 °C. IR (ATR) ν 3291 (NH), 1687 (C=O). ^1^H NMR (400 MHz, CDCl_3_) *δ* 1.45 (t, 3H, *J* = 7.1 Hz), 4.40 (dq, 1H, *J* = 10.9, 7.1 Hz), 4.46 (dq, 1H, *J* = 10.9, 7.1 Hz), 7.23 (dq, 1H, *J* = 2.8, 1.5 Hz), 7.25 (d, 1H, *J* = 2.5 Hz), 7.28–7.34 (m, 3H), 7.51–7.57 (m, 2H), 8.49 (d, 1H, *J* = 2.5 Hz), 9.50 (s, 1H); ^19^F NMR (376 MHz, CDCl_3_) *δ* 84.3 (s, CF_3_); ^13^C NMR (126 MHz, CDCl_3_) *δ* 14.3, 61.5, 82.0 (q, ^2^*J*_CF_ = 31.6 Hz), 117.1, 117.9, 118.4, 119.0, 121.4, 123.7 (q, ^1^*J*_CF_ = 283.6 Hz), 123.8, 126.0, 127.5, 128.0 (2C), 128.2 (2C), 129.0, 129.5, 134.2, 145.3, 159.9. HRMS (ESI) *m*/*z*: [M + H]^+^ calcd for C_21_H_15_Cl_2_F_3_NO_3_ 456.0376, found 456.0374.

*Ethyl 6,8-dibromo-4-phenyl-4-(trifluoromethyl)-2,4-dihydrochromeno[3,4-c]pyrrole-1-carboxylate* (**10dg**). Yield 191 mg (70%), mp 181–183 °C. IR (ATR) ν 3246 (NH), 1683 (C=O). ^1^H NMR (400 MHz, CDCl_3_) *δ* 1.46 (t, 3H, *J* = 7.1 Hz), 4.41 (dq, 1H, *J* = 10.8, 7.1 Hz), 4.46 (dq, 1H, *J* = 10.8, 7.1 Hz), 7.22 (dq, 1H, *J* = 2.8, 1.5 Hz), 7.28–7.35 (m, 3H), 7.52–7.60 (m, 3H), 8.67 (d, *J* = 2.3 Hz, 1H), 9.52 (s, 1H); ^19^F NMR (376 MHz, CDCl_3_) *δ* 84.3 (s, CF_3_); ^13^C NMR (126 MHz, CDCl_3_) *δ* 14.4, 61.5, 82.2 (q, ^2^*J*_CF_ = 31.7 Hz), 112.8, 115.0, 117.0, 117.9, 118.4, 118.9, 121.8, 123.7 (q, ^1^*J*_CF_ = 283.6 Hz), 128.1 (2C), 128.2 (2C), 129.4, 129.5, 134.2, 134.5, 146.8, 160.0. HRMS (ESI) *m*/*z*: [M + H]^+^ calcd for C_21_H_15_Br_2_F_3_NO_3_ 543.9365, found 543.9365.

### 3.3. Synthesis of Compounds ***13**–**15***

*Ethyl 2-methyl-4-(trifluoromethyl)-2,4-dihydrochromeno[3,4-c]pyrrole-1-carboxylate* (**13**). A mixture of 2,4-dihydrochromeno[3,4-c]pyrrole **12aa** (156 mg, 0.5 mmol), iodomethane (106 mg, 1.5 mmol) and K_2_CO_3_ (138 mg, 1.0 mmol) in acetone (3 mL) was heated at 40 °C for 18 h with stirring. Upon completion of the reaction, the mixture was poured into ice water (25 mL) and the precipitate was filtered and washed with water (10 × 5 mL). Yield 124 mg (76%), white powder, mp 68–70 °C. IR (ATR) ν 1690 (C=O). ^1^H NMR (400 MHz, CDCl_3_) *δ* 1.43 (t, 3H, *J* = 7.1 Hz), 3.91 (s, 3H), 4.42 (q, 2H, *J* = 7.1 Hz), 5.47 (q, 1H, *J* = 6.7 Hz), 6.76 (s, 1H), 6.99–7.06 (m, 2H), 7.15–7.21 (ddd, 1H, *J* = 8.1, 7.8, 1.4 Hz), 8.28 (dd, 1H, *J* = 7.8, 1.4 Hz); ^19^F NMR (376 MHz, CDCl_3_) *δ* 83.2 (d, *J* = 6.7 Hz, CF_3_); ^13^C NMR (126 MHz, CDCl_3_) *δ* 14.2, 38.5, 60.8, 70.9 (q, ^2^*J*_CF_ = 34.0 Hz), 110.2, 117.1, 118.6, 118.7, 121.7, 122.2, 123.4 (q, ^1^*J*_CF_ = 283.6 Hz), 124.2, 127.0, 128.6, 151.8, 161.4. HRMS (ESI) *m*/*z*: [M + H]^+^ calcd for C_16_H_15_F_3_NO_3_ 326.0999, found 326.1004.

*Ethyl 2-phenyl-4-(trifluoromethyl)-2,4-dihydrochromeno[3,4-c]pyrrole-1-carboxylate* (**14**). A mixture of chromene **10aa** (138 mg, 0,5 mmol), phenylboronic acid (122 mg, 1.0 mmol), Et_3_N (139 μL, 101 mg 1.0 mmol) and Cu(OAc)_2_ (91 mg, 0.5 mmol) in 1,2-dichloroethane (3 mL) was refluxed with stirring for 20 h. Upon completion of the reaction, the residue was evaporated under reduced pressure to complete dryness. The residue was purified by silica gel column chromatography (eluent EtOAc–hexane (1:3)). Yield 87 mg (45%), white powder, mp 115–116 °C. IR (ATR) ν 1682 (C=O). ^1^H NMR (400 MHz, CDCl_3_) *δ* 0.92 (t, 3H, *J* = 7.1 Hz), 4.08 (q, 2H, *J* = 7.1 Hz), 5.57 (qd, 1H, *J* = 6.6, 0.5 Hz), 6.92 (s, 1H), 7.03–7.09 (m, 2H), 7.22 (ddd, 1H, *J* = 8.1, 7.5, 1.6 Hz), 7.29–7.34 (m, 2H), 7.39–7.48 (m, 3H), 8.39–8.43 (m, 1H); ^19^F NMR (376 MHz, CDCl_3_) *δ* 83.3 (d, *J* = 6.6 Hz, CF_3_); ^13^C NMR (126 MHz, CDCl_3_) *δ* 13.5, 60.7, 70.9 (q, ^2^*J*_CF_ = 34.1 Hz), 111.2, 117.1, 118.0, 119.6, 122.0, 122.5, 123.4 (q, ^1^*J*_CF_ = 283.6 Hz), 123.5, 125.6 (2C), 126.8, 128.0, 129.0 (2C), 129.1, 141.0, 151.7, 161.3. HRMS (ESI) *m*/*z*: [M + H]^+^ calcd for C_21_H_17_F_3_NO_3_ 388.1155, found 388.1160.

*Ethyl 3-bromo-4-(trifluoromethyl)-2,4-dihydrochromeno[3,4-c]pyrrole-1-carboxylate* (**15**). To a solution of pyrrole **3aa** (0.5 mmol) cooled to 0–5 °C in CHCl_3_ (5 mL), NBS (98 mg, 0.55 mmol) was added portionwise within 25 min with stirring. Then, the reaction mixture was stirred overnight and the precipitate was filtered and washed with CHCl_3_ (1 mL). The solution was evaporated under reduced pressure to complete dryness, washed with water (3 × 5 mL), and recrystallized from 70% ethanol. Yield 107 mg (55%), white powder, mp 165–167 °C. IR (ATR) ν 3225 (NH), 1665 (C=O). ^1^H NMR (400 MHz, CDCl_3_) *δ* 1.44 (t, 3H, *J* = 7.1 Hz), 4.42 (dq, 1H, *J* = 11.6, 7.1 Hz), 4.45 (dq, 1H, *J* = 11.6, 7.1 Hz), 5.44 (q, 1H, *J* = 6.7 Hz), 7.03–7.09 (m, 2H), 7.24 (td, 1H, *J* = 7.8, 1.5 Hz), 8.62 (dd, 1H, *J* = 7.8, 1.5 Hz), 9.42 (s, 1H); ^19^F NMR (376 MHz, CDCl_3_) *δ* 83. 8 (d, *J* = 6.7 Hz, CF_3_); ^13^C NMR (126 MHz, CDCl_3_) *δ* 14.4, 61.3, 69.8 (q, ^2^*J*_CF_ = 34.1 Hz), 102.4, 112.1, 117.0, 117.1, 118.5, 122.6 (2C), 123.5 (q, ^1^*J*_CF_ = 286.7 Hz), 127.8, 129.8, 151.8, 159.3. HRMS (ESI) *m*/*z*: [M + H]^+^ calcd for C_15_H_12_BrF_3_NO_3_ 389.9947, found 389.9955.

## Data Availability

Data are contained within the article and Appendix A.

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
