# Peer review of "Green and Efficient Construction of Chromeno[3,4-c]pyrrole Core via Barton–Zard Reaction from 3-Nitro-2H-chromenes and Ethyl Isocyanoacetate"

_molecules, 2022, doi:10.3390/molecules27238456_

Round 1
Reviewer 1 Report
This manuscript describes an efficient regioselective route to 2,4-dihydrochromeno[3,4-c]pyrroles by the Barton-Zard reaction of 3-nitro-2H-chromenes and ethyl 3-isocyanoacetate. A relatively broad substrate scope was demonstrated, and the products were prepared in good to high yields. The reaction is scalable and operates under green conditions (ethanol as the solvent and K2CO3 as the base), which is an additional benefit for practical use. The obtained chromeno[3,4-c]pyrrole derivatives are of interest for medicinal chemistry applications. Also, derivatization of the core structure in these products can be easily performed as demonstrated by the authors. The manuscript can be recommended for publication in Molecules after a minor revision:
1) Are chromenes 10 containing a nitro-group in the benzene tolerated in the synthesis of pyrroles 12?
2) Atom numbering should be added in the general formula of products 12 given in Scheme 3.
3) Page 4, line 92. “DABSO” (should be “DABCO”)
4) Page 7, line 131. “26-3.7 Hz” (should be “2.6-3.7 Hz”)
Reviewer 2 Report
Dear Editor,
Please find attached comments on Manuscript ID: molecules-2059303, entitled “Green and efficient construction of chromeno[3,4-c]pyrrole core 2 via Barton-Zard reaction from 3-nitro-2H-chromenes and ethyl 3-isocyanoacetate”, Corresponding Authors: Vladislav Y. Korotaev and Vyacheslav Y. Sosnovskikh
The manuscript describes the synthesis of a series of chromeno[3,4-c]pyrrole derivatives.
Overall, the synthesis of the new compounds is carried out by 1 step reaction and was elaborated precisely and detailed. The purity of the final compounds was confirmed by thin layer chromatography (TLC), melting point determination, IR (ATR) , HRMS, 1H‐NMR, 13C‐NMR and 19F NMR.
The authors should provide in the Supplementary material 19F NMR spectra, IR (ATR) and HRMS.
In general, the paper needs some attention to the typos before publication.
In summary, the manuscript is interesting and provides a synthetic foundation for future efforts to developing of antibacterial agents.
After a minor revision, it would be suitable for publication in Molecules.
Reviewer 3 Report
The paper is well written and needs only a few correction and completion, mainly concerning the literature which was used for synthetic methods and structure elucidation.
For more details see attached file.
